# Ultrasonic Microbubble Cavitation Deliver Gal-3 shRNA to Inhibit Myocardial Fibrosis after Myocardial Infarction

**DOI:** 10.3390/pharmaceutics15030729

**Published:** 2023-02-22

**Authors:** Wenqu Li, Qiaofeng Jin, Li Zhang, Shukun He, Yishu Song, Lingling Xu, Cheng Deng, Lufang Wang, Xiaojuan Qin, Mingxing Xie

**Affiliations:** 1Department of Ultrasound Medicine, Union Hospital, Tongji Medical College, Huazhong University of Science and Technology, 1277 Jiefang Ave., Wuhan 430022, China; 2Hubei Province Clinical Research Center for Medical Imaging, Wuhan 430022, China; 3Hubei Province Key Laboratory of Molecular Imaging, Wuhan 430022, China

**Keywords:** ultrasonography, myocardial fibrosis, microbubbles, Galectin-3

## Abstract

Galectin-3 (Gal-3) participates in myocardial fibrosis (MF) in a variety of ways. Inhibiting the expression of Gal-3 can effectively interfere with MF. This study aimed to explore the value of Gal-3 short hairpin RNA (shRNA) transfection mediated by ultrasound-targeted microbubble destruction (UTMD) in anti-myocardial fibrosis and its mechanism. A rat model of myocardial infarction (MI) was established and randomly divided into control and Gal-3 shRNA/cationic microbubbles + ultrasound (Gal-3 shRNA/CMBs + US) groups. Echocardiography measured the left ventricular ejection fraction (LVEF) weekly, and the heart was harvested to analyze fibrosis, Gal-3, and collagen expression. LVEF in the Gal-3 shRNA/CMB + US group was improved compared with the control group. On day 21, the myocardial Gal-3 expression decreased in the Gal-3 shRNA/CMBs + US group. Furthermore, the proportion of the myocardial fibrosis area in the Gal-3 shRNA/CMBs + US group was 6.9 ± 0.41% lower than in the control group. After inhibition of Gal-3, there was a downregulation in collagen production (collagen I and III), and the ratio of Col I/Col III decreased. In conclusion, UTMD-mediated Gal-3 shRNA transfection can effectively silence the expression of Gal-3 in myocardial tissue, reduce myocardial fibrosis, and protect the cardiac ejection function.

## 1. Introduction

Myocardial fibrosis (MF) is the main pathological feature and the leading cause of irreversible chronic heart failure, which is a serious threat to life. MF is a tissue repair mechanism caused by a progressive accumulation of extracellular matrix in response to injury, inflammation or stress [1]. The process leads to impaired tissue repair, causing tissue and organ scarring, resulting in myocardial tissue remodeling and reduced or lost heart function [2,3]. Inhibiting MF is helpful for protecting heart function. At present, there is still a lack of effective means to intervene in fibrosis [4]. Gene-targeted therapy has become a new hope, but the main limiting factor is the lack of ideal targets for MF.

Galectin-3 (Gal-3) is a shared factor in tissue fibrosis, especially in MF [5,6]. Several reports have suggested that Gal-3 upregulation aggravates myocardial inflammation and promotes fibroblast differentiation, cardiac myofibroblast activation, and collagen maturation. Therefore, Gal-3 inhibition has been considered a potential target in curing MF [7,8]. Preclinical studies have reported that pharmacological and genetic inhibition of Gal-3 confers effects with reduction of MF [9,10]. Modified citrus pectin (MCP) and N-acetyllactosamine, inhibitors of Gal-3, ameliorated MF and inflammation in various rodent and mice models [10]. Sharma, U. C et al. also reported that cardioselective Gal-3 overexpression mediates cardiac fibro-inflammatory response and heart failure. N-acetyl-seryl-aspartyl-lysyl-proline, an endogenous peptide, can reverse the fibro-inflammatory cascade triggered by Gal-3 [11]. While the anti-fibrosis effects of Gal-3 on animal experiments were viewed as beneficial, current drugs targeting Gal-3 on patients showed less effectiveness. Lau, E. S. et al., who examined the effect of Gal-3 inhibition with MCP in patients with hypertension and elevated Gal-3 levels, observed no changes in collagen markers after MCP therapy [12]. Although inhibition of Gal-3 expression has become a promising precision-based strategy to reduce fibrosis and prevent left ventricular remodeling, how to effectively inhibit Gal-3 in the heart remains a problem in translational studies.

The current treatment strategies targeting Gal-3 have two main problems. Most studies use carbohydrate-based compounds as Gal-3 blockers. However, these drugs counteract Gal-3 activity through the carbohydrate-binding domain, which blocks only part of Gal-3 function [13]. On the other hand, the current treatment lacks targeting. All body tissues can be exposed to drugs, which increase the side effects of therapy. In the current study, we use a Gal-3 shRNA to block the translation of Gal-3, which can suppress Gal-3 expression at the gene level. Ultrasound-targeted microbubble destruction (UTMD), a gene delivery method with high repeatability, effective gene transfection, targeted delivery and low pathogenicity, has been applied to deliver shRNA to the infarcted myocardium [14,15]. Therefore, we used UTMD to transfect Gal-3 shRNA to suppress Gal-3 expression in the infarcted heart and achieve the purpose of inhibiting MF and protecting heart function, as shown in Figure 1.

## 2. Materials and Methods

### 2.1. Plasmid Construction

Gal-3 shRNA knockdown constructs were inserted into the hU6-CMV-Puro-GFP(Gv493) vector (Genechem, Shanghai, China). Three Gal-3 shRNA constructs were transfected into H9C2 cells using lipo2000 (Invitrogen, Waltham, MA, USA) according to the manufacturer’s instructions. We then screened the protein expression of Gal-3 for a kind of plasmid with the powerful Gal-3 knockdown efficiency.

### 2.2. Gal-3 shRNA Loaded Cationic Microbubbles (Gal-3 shRNA/CMBs) Preparation and Characterization

Cationic microbubbles (CMBs) were synthesized using the thin-film hydration method as previously described. Briefly, 1,2-distearoyl-sn-glycero-3-phosphocholine (DSPC) (Avanti Polar Lipids, Alabaster, AL, USA), 1,2-distearoyl-sn-glycero-3-phosphoethanolamine-N- [methoxy(polyethyleneglycol)-2000] (DSPE-PEG2000) (Corden Pharma Inc., Liestal, Switzerland) and 1,2-stearoyl-3-trimethylammonium-propane (DSTAP) (Avanti Polar Lipids, Alabaster, AL, USA) were dissolved in chloroform and mixed at a molar ratio of 7.5:0.5:2. The solution was dried with nitrogen and resuspended at 3 mg/mL in 0.1 M Tris buffered saline (with 10% glycerol and 10% propylene glycol), followed by sonication to disperse the lipids. The solution was placed in a sealed glass vial (1 mL each), and the headspace was filled with perfluoropropane gas. Then we used an agitator to shake for 30 s to form CMBs. Finally, 120 μg Gal-3 shRNA and 120 μL CMBs were mixed and resuspended in a 500 μL PBS solution and incubated at room temperature for 20 min to form Gal-3 shRNA/CMBs via charge–charge interactions.

The appearance of Gal-3 shRNA/CMBs was observed under a microscope (OLYMPUS IX73, Tokyo, Japan). The surface potential and particle size of Gal-3 shRNA/CMBs were measured by Zeta PALS potential and particle size analyzer (Brookhaven Instruments Corporation, Holtsville, NY, USA), and the binding rate between Gal-3 shRNA and CMBs was measured by flow cytometry analysis (BD, Franklin Lakes, NJ, USA).

### 2.3. Myocardial Infarct (MI) Animal Models

Male SD rats (180~230 g) were purchased from Beiente Biotechnology Co., Ltd. (Wuhan, China). All animal care and experiments were conducted in accordance with the institutional guidelines of the Animal Care and Use Committee of Huazhong University of Science and Technology, Wuhan, China. Rats were anesthetized with a single dose of pentobarbital sodium (60 mg/kg, intraperitoneally) and kept breathing via tracheal intubations connected to an animal ventilator. The MI rat models were established by ligating the left anterior descending coronary artery. The left anterior chest was cut to expose the heart, and the left anterior descending coronary artery was identified and permanently ligated with a 6–0 silk thread. Whiting the left ventricular anterior wall indicated the MI model was successful, and the sternal was closed.

### 2.4. Treatment of MI Rats

The models were randomized to the Gal-3 shRNA/CMBs + ultrasound (Gal-3 shRNA/CMBs + US) group and the control group. The Gal-3 shRNA/CMBs + US group was given the treatment of uniformly injecting 500 μL Gal-3 shRNA/CMBs solution within 20 min. At the same time, ultrasound irradiation (Sonitron 2000 V, NEPA GENE, Ichikawa, Japan) with 1 MHz, 50% duty cycle and 2 W/cm^2^ was given to the precordial area, and the irradiation time was 20 min [16]. The control group was treated similarly, but the injecting drugs were changed to the PBS solution. All treatment was given on post-operation days 1, 3, and 7.

### 2.5. Echocardiography In Vivo

Echocardiography with GE vivid E95 (12S, GE healthy, Chicago, IL, USA) was performed during preoperation and on post-operation days 7, 14 and 21. The left ventricular ejection fraction (LVEF), fractional shortening (FS), left ventricular internal diameter at end-systole (LVIDs), and left ventricular internal diameter at end-diastole (LVIDd) were measured by M-mode images at the papillary muscle level short-axis, as shown in Appendix A.

### 2.6. Histological Analysis

For histological analysis, heart tissue was divided into infarcted area, border zone, and non-infarcted area, as shown in Appendix A. Hearts were fixed in 4% formaldehyde, embedded by paraffin and sectioned. Haematoxylin and eosin (HE) staining was used to assess the morphological structure. A Masson Trichrome Stain kit was used to analyze the myocardial fibrosis degree.

### 2.7. Western Blotting

The heart sample was homogenized using metal beads in RIPA buffer with protease inhibitor (Beyotimem, Shanghai, China) and PMSF (Beyotime, Shanghai, China). The concentration of total protein was determined by the BCA (Boster Biological Technology, Wuhan, China) method. A 5× loading buffer (Beyotime, Shanghai, China) was added to the sample at the ratio of 4:1 and boiled for 5 min at 95 °C. Then, 30 μg protein of each sample was run on a 12% SDS-PAGE gel (Epizyme Biomedical Technology, Shanghai, China), transferred to PVDF membranes (Boster Biological Technology, Wuhan, China) and then blocked with 5% non-fat milk for 60 min. The membranes were incubated with primary antibodies against Gal-3 (ABclonal, Wuhan, China) or GAPDH (ABclonal, Wuhan, China) at 4 °C overnight, and then membranes were washed with TBST and incubated with HRP-conjugated secondary antibodies (ABclonal, Wuhan, China) for 1 h. The blots were developed using the ECL chemiluminescence kit (Boster Biological Technology, Wuhan, China). The levels of target proteins were normalized to GAPDH.

### 2.8. Quantitative Polymerase Chain Reaction (qPCR)

Total RNA was isolated from myocardial tissue using the RNA isolation Total RNA Extraction Reagent (Vazyme Biotech, Shanghai, China) and reverse-transcribed into cDNA (1 μg total RNA per sample) using HiScript III RT SuperMix for qPCR(+gDNA wiper) (Vazyme Biotech, Shanghai, China) according to the manufacturer’s instructions. A NanoDrop 2000 Spectrophotometer was used to determine RNA concentrations. Quantitative real-time PCR analyses of the following genes were carried out with AceQ qPCR SYBR Green Master Mix (Vazyme Biotech, Shanghai, China). Data were analysed by the Bio-Rad CFX Connect system (Bio-Rad, Hercules, CA, USA). The mRNA expression levels were quantified using the 2^−ΔΔCt^ method. The primer sequences used for the reaction were as follows:
Gal-3-FGAACGACATCGCCTTCCACT(5′-3′)Gal-3-RCCCAGTTATTGTCCTGCTTCGCol1a1-FACAGCGTAGCCTACATGGCol1a1-RAAGTTCCGGTGTGACTCGCol1a2-FATGGTGGCAGCCAGTTTGCol1a2-RGCTGTTCTTGCAGTGGTAGGCol3a1-FTGGAAACCGGAGAAACATGCCol3a1-RCAGGATTGCCATAGCTGAACGAPDH-FACAGCAACAGGGTGGTGGACGAPDH-RTTTGAGGGTGCAGCGAACTT

### 2.9. Statistics

All data were expressed as the mean value ± standard deviation (mean ± SD). GraphPad Prism software was used for statistical analysis. Differences among groups were evaluated by analysis of variance or the unpaired Student’s *t* test. A *p* value < 0.05 was considered statistically significant. For the measurements of surface potential and particle size, the error bar was obtained from three independent samples, and each sample was measured twice. For the animal experiment, the error bar was obtained from three independent rats.

## 3. Results

### 3.1. Characterization of the Gal-3 shRNA/CMBs

As shown in Figure 2, the average particle size of CMBs and Gal-3 shRNA/CMBs were (1056 ± 64.5) nm and (3601 ± 371.2) nm, respectively. The mean Zeta potential of CMBs and Gal-3 shRNA/CMBs were (17.92 ± 2.796) mV and (−50.87 ± 5.057) mV, respectively. The binding rate of Gal-3 shRNA and CMBs was 67.4% according to flow quantitative analysis (Appendix A).

### 3.2. The Expression of Gal-3 after MI

In order to determine the optimal time point of Gal-3 intervention in MI rats, the expression level of Gal-3 and the degree of myocardial fibrosis were detected after MI. HE staining demonstrated that inflammatory infiltration remained the primary pathological feature in the first 3 days after ischemic cardiac injury. After that, the injured region of the hearts was gradually replaced by fibrous tissues. Compensatory myocardial hypertrophy occurred in the region remote from the area of injury, and left ventricular remodeling turned out to be obvious with the time prolonging. With ischemic time prolonging, the area of fibrotic scar tissue was significantly greater according to the result of Masson staining (Figure 3B). Gal-3 increased within 7 days, then declined by day 21, and began to rise again in day 28 after MI (Figure 3C and Appendix A). Therefore, we chose day 1, 3 and 7 after MI to intervene Gal-3 expression in the heart of rats using UTMD-mediated delivery. Compared to the control group, the Gal-3 mRNA in Gal-3 shRNA/CMBs + US group showed a downward trend on day 7 and was significantly lower in three regions of heart on day 21 (Figure 3D). Correspondingly, on day 7, there was no significant difference in Gal-3 protein between the control and the Gal-3 shRNA/CMBs + US groups. On day 21, Gal-3 protein was lower in the infarcted area and border zone (*p* < 0.05), but there was no significant difference in the non-infarcted area (*p* > 0.05), as shown in Figure 3E. Immunohistochemistry staining also indicated that fewer Gal-3+ cells infiltrated to the myocardium in the Gal-3 shRNA/CMBs + US group (Figure 3F). The serum Gal-3 level displays a slight decrease only on day 7 (Figure 3G).

### 3.3. Cardiac Function

Then we used M-mode echocardiography to evaluate the change of cardiac function in each group. We chose LVEF, FS as main indicators of cardiac systolic function, which remained high in normal heart and decreased in damaged heart. In addition, we chose LVIDd and LVIDs as main indicators of cardiac chamber size, which increased when cardiac function was poor. As shown in Figure 4A, in the control group, the motion of the anterior wall of the heart gradually decreased with ischemic time prolonging. However, in the Gal-3 shRNA/CMBs + US group, the anterior wall still retained a weak motion until day 21. On the day 7 after MI, compared to the control group, the LVEF, FS, LVIDd, and LVIDs of the Gal-3 shRNA/CMBs + US group had no significant difference (60.11 ± 7.38% vs. 56.27 ± 10.59%, 27.78 ± 3.60% vs. 20.99 ± 6.15%, 6.99 ± 1.27 cm vs. 7.22 ± 0.92 cm; 4.88 ± 1.42 vs. 5.53 ± 0.87, *p* > 0.05). On day 14, the values of LVEF and FS in the Gal-3 shRNA/CMBs + US group were higher than that of the control group (67.56 ± 2.58% vs. 44.48 ± 7.69%, 33.25 ± 1.35% vs. 21.39 ± 5.39%, *p* < 0.01, respectively); however, the LVIDd and LVIDs between the two groups had no significant difference (6.64 ± 0.57 cm vs. 7.45 ± 1.17 cm, 4.28 ± 0.33 cm vs. 5.67 ± 1.10 cm, *p* > 0.05). On day 21, compared with the control group, the EF and FS were higher (67.68 ± 2.03% vs. 46.87 ± 3.44%, 33.49 ± 2.35% vs. 20.47 ± 4.45%, *p* < 0.01, respectively), and the values of LVIDd and LVIDs were smaller in the Gal-3 shRNA/CMBs + US group (6.64 ± 0.57 cm vs. 8.73 ± 1.06 cm, 4.64 ± 0.78 cm vs. 6.86 ± 1.23 cm, *p* > 0.05, respectively), as shown in Figure 4B,C.

### 3.4. Myocardial Fibrosis and Macrophage Infiltration

As shown in Figure 5A, the scar area of the two groups showed no significance on day 7, but the scar area in the Gal-3 shRNA/CMBs + US group was reduced by 5% compared with the control group on day 21. In three regions of the left ventricle, the degree of fibrosis in the Gal-3 shRNA/CMBs + US group was lower than that of the control group (Figure 5B). Moreover, there was less macrophage infiltration in the Gal-3 shRNA/CMBs + US group (Figure 5C).

### 3.5. The mRNA Levels of Collagen

Collagen expression on day 21 is shown in Figure 6. COL1A1 and COL3A1 (the critical transcriptional factor for collagen I and III) of the Gal-3 shRNA/CMBs + US group were lower than those of the control group in the infarcted area and border zone (*p* < 0.05), but there was no significant difference in the non-infarcted area (*p* > 0.05). Nevertheless, Col I/Col III of the Gal-3 shRNA/CMBs + US group were lower than those of the control group only in the infarcted region (*p* < 0.05), and no significant difference was observed in the other two regions (*p* > 0.05), as shown in Figure 6.

### 3.6. Toxicity Evaluation

In order to evaluate the toxicity, we use HE staining to evaluate morphological changes in major tissues and organs. Additionally, a serum index was detected to evaluate liver and kidney function. As shown in Appendix A, the two groups’ morphology and structure of the lung, liver, spleen, and kidney were normal without inflammatory cell infiltration. After the treatment of Gal-3 shRNA/CMBs + US, serum levels of ALT, AST, Urea Crea and the main indexes in the examination of blood routine were within the normal range, and there was no difference between the two groups (*p* > 0.05) (Appendix A).

## 4. Discussion

Myocardial fibrosis is an important cause of irreversible impairment of cardiac function. Preventing fibrosis can effectively protect heart function [17]. Gal-3 is an ideal target for the treatment of fibrosis, and previous studies have demonstrated that the Gal-3 inhibitor can effectively interfere with fibrosis in mice receiving adrenergic agonists or transaortic constriction [10,18]. Some Gal-3 inhibitors have already entered into clinical trials. Belapectin, an inhibitor of Gal-3, has been applied in patients with nonalcoholic steatohepatitis with cirrhosis and portal hypertension. Though belapectin did reduce the hepatic venous pressure gradient and development of varices in a subgroup analysis of patients without esophageal varices, it is disappointing that belapectin did not exhibit robust efficacy in the improvement of fibrosis [19]. Besides, another Gal-3 inhibitor, MCP, has been used in patients with hypertension and elevated Gal-3 levels. However, MCP also did not reach ideal effects. There were no changes observed in collagen markers after MCP therapy, and some patients had severe gastrointestinal side effects [12]. Therefore, it is necessary to further explore new ways to inhibit the expression of Gal-3. In this study, we demonstrated a new method for targeted inhibition of Gal-3 in MI rats by ultrasonic microbubble cavitation to mediate Gal-3 shRNA delivery. Our result proved that Gal-3 shRNA transfection could effectively decrease Gal-3 expression in the heart and reduce myocardial fibrosis caused by excessive Gal-3 secretion in the late stage of MI.

### 4.1. UTMD-Mediated Gal-3 shRNA Transfection in Myocardial Tissue

The major drawback of nucleic acid drugs is their rapid clearance by nucleases in vivo. Therefore, a delivery carrier is needed for in vivo application. Using phospholipid microbubbles to load nucleic acid drugs is an ideal strategy for improving their stability in the blood [20]. Microbubbles have reportedly been able to load nucleic acids through electrostatic interaction, avidin–biotin interaction and the nucleic acids-encapsulation method, in which electrostatic interaction is a simple stabilization method to synthesize nucleic-acid-loaded microbubbles [21]. In this study, we use electrostatic interaction to prepare Gal-3 shRNA/CMBs, and the zeta potential of the microbubbles was reversed to negative after the CMBs and shRNA were mixed, which indicated that Gal-3 shRNA was adsorbed on the surface of the microbubbles. The flow cytometry further confirmed that the binding rate of CMB and Gal-3 shRNA was 67.4%, as shown in Figure 2.

Most nucleic acid drugs have difficulty passing through the plasma membrane due to their ionic characteristics and high molecular weight. Thus, nucleic acid drugs have a low transfection rate [22]. UTMD can be used to increase the permeability of biological barriers and enhance the gene transfection rate through its cavitation and sonoporation effect [23]. Moreover, ultrasound irradiation can trigger the release of copious nucleic acid from microbubbles, which increases the drug concentration in targeted sites [21,24]. Numerous studies have shown that UTMD-mediated gene delivery regressed myocardial hypertrophy, suppressed inflammatory responses, and protected ischemia–reperfusion injury [25,26,27]. In this study, as shown in Figure 3 and Figure 5, Gal-3 mRNA and protein expression of myocardial tissue were reduced after UTMD mediated Gal-3 shRNA transfection in the infarcted area and border zone, and the degree of fibrosis also decreased. These results proved that UTMD-mediated Gal-3 shRNA effectively inhibits Gal-3 expression at the transcription and translation level and reduces myocardial fibrosis. While coronary artery ligation resulted in a reduction of the blood supply in the infarcted area, from the second day after coronary artery ligation the capillary network in the border zone started expanding, with extensive branching and vessels sprouting into the infarcted area [28], which may contribute to the delivery of Gal-3 shRNA in the ischemic area. Besides, the permeability of infarcted area was increased due to the inflammatory reaction, which could be beneficial to Gal-3 shRNA delivery. In the non-infarcted area, the expression of Gal-3 mRNA and the degree of myocardial fibrosis was lower than that of the control group. Gal-3 protein expression of the Gal-3 shRNA/CMB + US group showed a down-regulated trend in the non-infarcted area, although the effect was not robust enough to reach statistical significance. Several reasons may contribute to this phenomenon. The detection time point is too early; unlike the infarcted area and border zone, the non-infarcted area has no apparent inflammatory reaction and neovascularization [28,29], which may reduce the transfection efficiency; meanwhile, the macrophages, the primary secretory cells of Gal-3 protein, were fewer in the non-infarct area than in the other two regions after MI [30]. Therefore, the inhibition effect in this region is not obvious.

### 4.2. Inhibition of Gal-3 Expression Can Reduce the Degree of Fibrosis

Myocardial fibrosis is characterized by excessive Col I and Col III deposition in extracellular matrix [31]. Col I is the major fibrillar component in extracellular matrix and determines myocardial stiffness. In contrast, Col III is synthesized by cardiac fibroblasts and is mainly responsible for myocardial elasticity [32,33]. In the late stage of MI, the Col I/Col III ratio increases, especially in the infarcted area due to the overexpression of Col I. The deregulated expression of the collagen ratio breaks the biomechanical balance of the collagen network, resulting in an increase in myocardial stiffness and the decline of LVEF [33,34]. Our data showed that Gal-3 shRNA transfection decreased the Col I and Col III mRNA levels in the infarcted area and border zone, while measures of Col I and Col III in the non-infarcted area were not affected following treatment, as shown in Figure 6. This is consistent with the results of gal-3 protein inhibition, which indicated that UTMD-mediated Gal-3 shRNA transfection reduced the Gal-3 expression in vivo. Gal-3 inhibition led to the depression of collagen production and further inhibited myocardial fibrosis. In addition, the Col I/Col III ratio was reduced after the Gal-3 inhibition in the infarcted area, resulting in a decrease in myocardial stiffness and protective LVEF.

### 4.3. Inhibiting Fibrosis Can Preserve Cardiac Function

Accumulating evidence suggested that Gal-3 played a prominent role in the pathogenesis of MF [11]. Gal-3 inhibition conferred cardioprotective effects by suppressing MF [35]. In the current study, we also found that LVEF and FS were improved after Gal-3 inhibition, as shown in Figure 4. This result demonstrated that UTMD-mediated Gal-3 shRNA transfection could decrease MF and protect LVEF. Besides, phenotypes associated with Gal-3 overexpression, such as left ventricular dilatation, were attenuated by Gal-3 shRNA transfection.

### 4.4. Study Limitation

The mean limitation of this study is that the observation time is short. In the present study, we only detected the Gal-3 level and MF on day 21 after MI and did not observe the duration of the Gal-3 inhibition effect and the impact on prognosis. In addition, we did not conduct a thorough study on the mechanism of the anti-myocardial fibrosis by Gal-3 shRNA transfection.

## 5. Conclusions

In the present study, we applied ultrasonic microbubble cavitation to deliver Gal-3 shRNA to transfect a rat MI model. The results showed that the scar area in the Gal-3 shRNA/CMBs + US group was reduced by 5% from that in the control group, and LVEF in Gal-3 shRNA/CMBs + US group was improved by 20% over that in the control group. This indicated ultrasonic microbubble cavitation delivery of Gal-3 shRNA can effectively silence the expression of Gal-3 in myocardial tissue, reduce MF and preserve cardiac function. We believe that our study is an attempt at targeted inhibition of Gal-3 and a decrease in myocardial fibrosis and that these data are useful for enhancing anti-fibrosis therapy development.

## Figures and Tables

**Figure 1 pharmaceutics-15-00729-f001:**
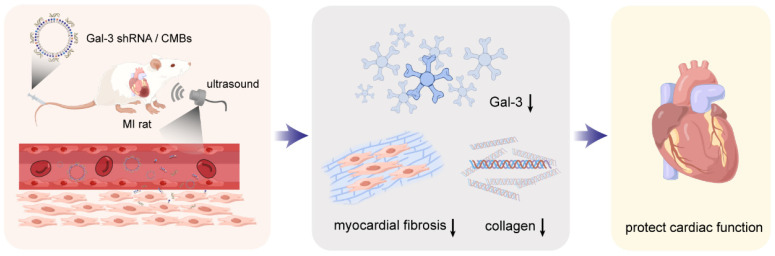
Schematic representation of Gal-3 shRNA delivery mediated cardiac function protection. The arrow represents downregulation (This figure was drawn by Figdraw).

**Figure 2 pharmaceutics-15-00729-f002:**
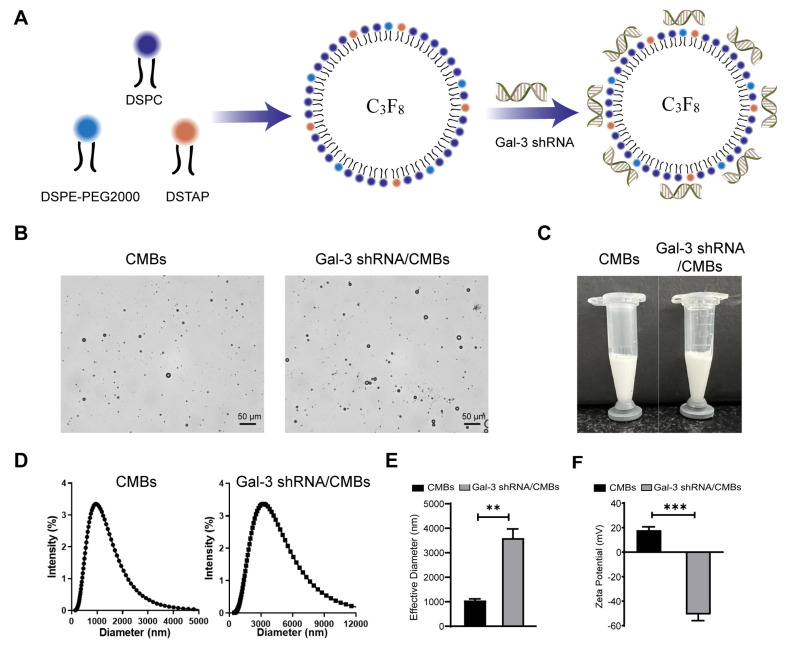
Characterization of Gal-3 shRNA/CMBs. (**A**) Schematic representation of Gal-3 shRNA/CMBs preparation. (**B**) Representative optical micrographs of CMBs and Gal-3 shRNA/CMBs. (**C**) Appearance of the CMBs and Gal-3 shRNA/CMBs. (**D**) Size distribution of CMBs and Gal-3 shRNA/CMBs. (**E**) Average particle size of CMBs and Gal-3 shRNA/CMBs. (**F**) Average Zeta potentials of CMBs and Gal-3 shRNA/CMBs. (*n* = 3, ** *p* < 0.01 *** *p* < 0.001).

**Figure 3 pharmaceutics-15-00729-f003:**
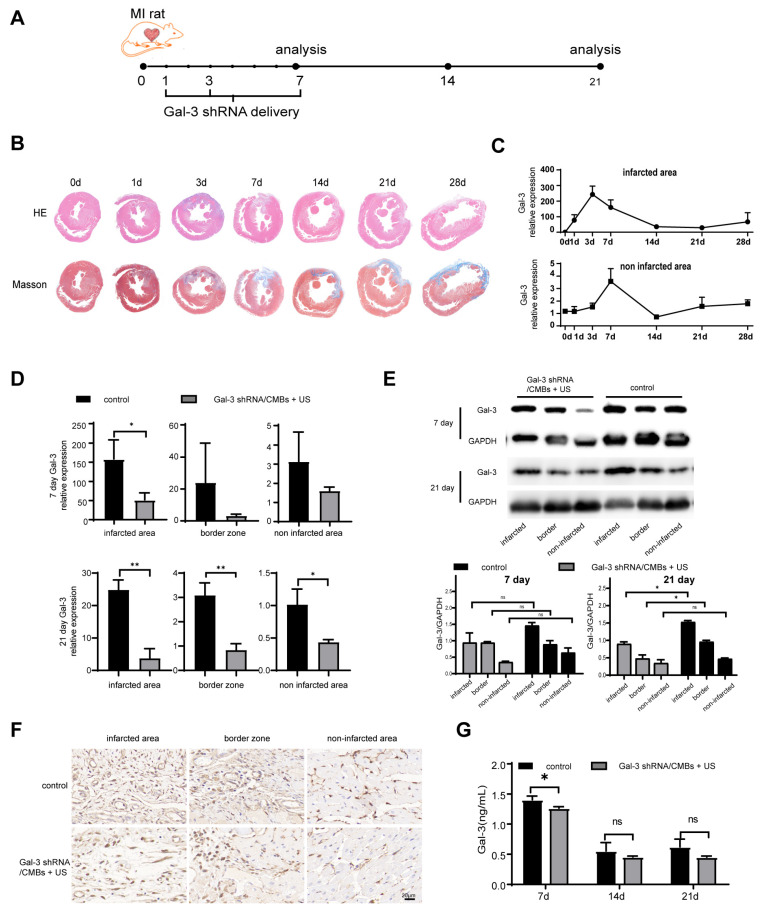
Gal-3 expression in myocardium. (**A**) Therapy regimen schematic of Ultrasonic Microbubble Cavitation-mediated Gal-3 shRNA delivery in MI rats. (**B**) HE staining and Masson trichrome staining of hearts sectioned at mid-ventricle at 0, 1, 3, 7, 14, 21, 28 days post-MI. (**C**) Temporal changes in Gal-3 mRNA expression post-MI. (**D**) Myocardial Gal-3 mRNA level in Gal-3 shRNA/CMBs + US group and control group. (**E**) Gal-3 protein expression in Gal-3 shRNA/CMBs + US group and control group. (**F**) Immunohistochemistry staining with Gal-3+ cells on day 21. (**G**) Concentration of Gal-3 in serum. Infarcted represents the infarcted area, border represents the border zone, and non-infarcted represents the non-infarcted area (*n* = 3, ns *p* > 0.05, * *p* < 0.05, ** *p* < 0.01).

**Figure 4 pharmaceutics-15-00729-f004:**
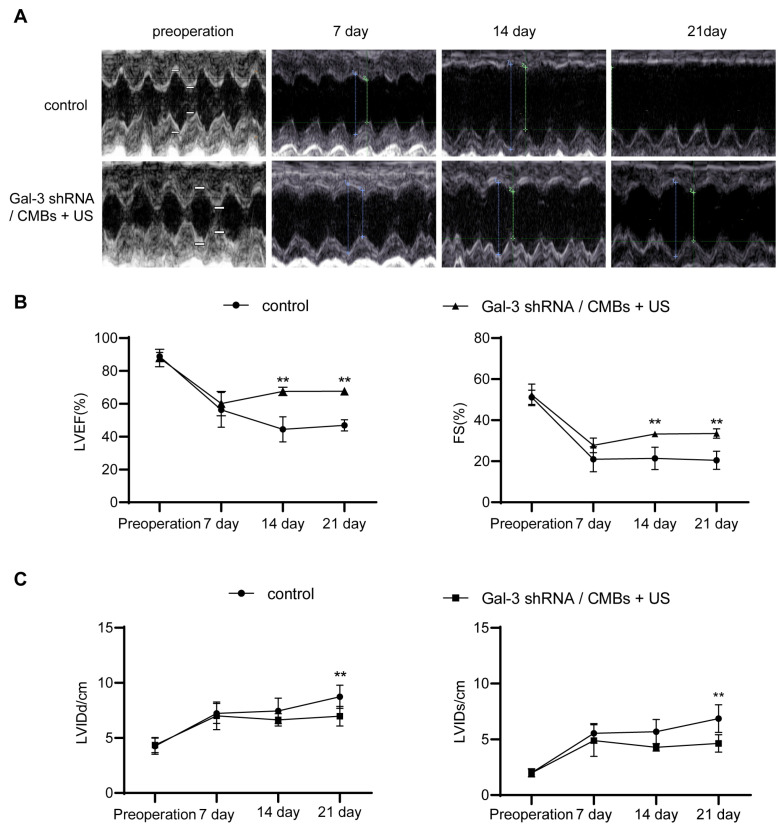
The improvement in cardiac function in MI rats. (**A**) Representative M-mode echocardiography images. The symbol showed the measurement of cardiac function. (**B**,**C**) Cardiac function indicators of LVEF, FS, LVIDd and LVIDs were evaluated accordingly. (*n* = 3, ** *p* < 0.01).

**Figure 5 pharmaceutics-15-00729-f005:**
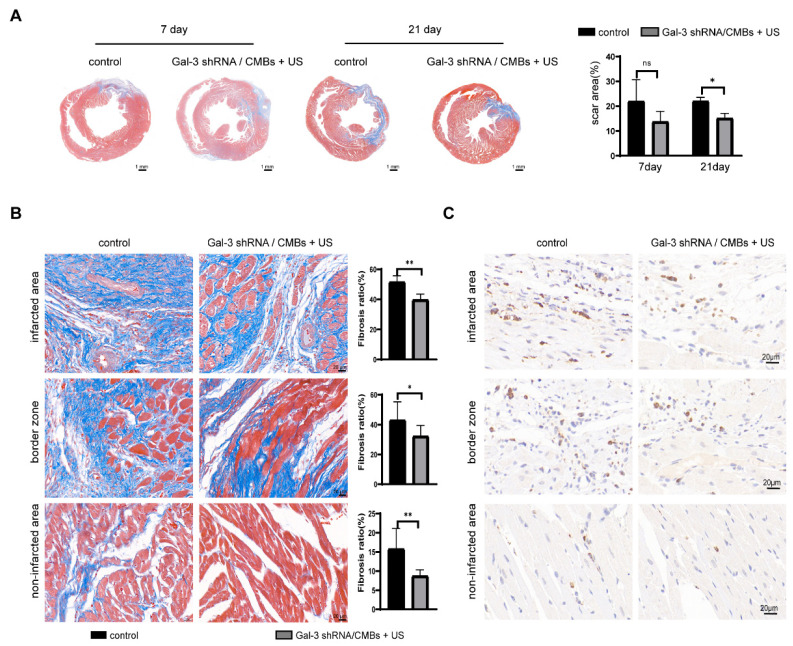
Myocardial fibrosis and macrophage infiltration in MI rats. The blue-stained area represents the fibrous tissue, and the red-stained area represents the myocardial tissue. (**A**) Masson staining and quantitative analysis of scar area in the two groups on day 7 and day 21. (**B**) Degree of fibrosis and quantitative analysis of the fibrosis ratio in three regions between the two groups on day 21 (**C**) Immunohistochemistry staining with CD68+ cells on day 21 (*n* = 3, ns *p* > 0.05, * *p* < 0.05, ** *p* < 0.01).

**Figure 6 pharmaceutics-15-00729-f006:**
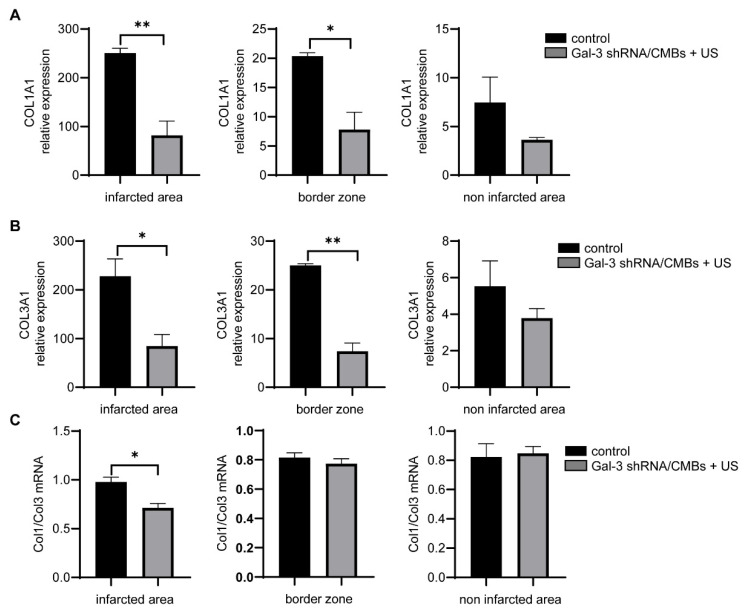
Collagen synthesis in MI rats. (**A**) COL1A1 mRNA expression in different regions of MI heart. (**B**) COL3A1 mRNA expression in different regions of MI heart. (**C**) Ratio of Col1/Col3 mRNA expression in different regions of MI heart (*n* = 3, * *p* < 0.05, ** *p* < 0.01).

## Data Availability

Not applicable.

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
