# Peer review of "Ultrasonic Microbubble Cavitation Deliver Gal-3 shRNA to Inhibit Myocardial Fibrosis after Myocardial Infarction"

_pharmaceutics, 2023, doi:10.3390/pharmaceutics15030729_

Round 1

Reviewer 1 Report

The manuscript approaches the “forever hot topic” of the remodeling process after myocardial infarction (MI), and re-introduces the Galactin-3 as a potential target in benefit of healing after MI.

1.       The authors performed histological staining of heart tissue at different time-points after MI and they show representative images in Figure 3B and Figure 5B.

After the permanent left anterior descending coronary artery (LAD) ligation, the scar tissue could slightly expand with the time, but usually the most changes don’t come from the scar tissue expansion (excepting late heart failure) but from it’s remodeling (composition of inflammatory/reparatory cells, fibroblasts and extracellular matrix, and the development of neovasculature).

a)       How do the authors explain the enlargement of the scar area from day 7 to day 21 with a regression at day 14 (Figure 3B)?

b)      How do the authors explain the difference in scar area size  within the same group (14days post MI) when comparing the histological slides presented in Figure 3B and Figure 5B, since in Figure 5B the scar area 14days post MI seems to be a lot smaller than in Figure 3B 14 days post MI?

2.       Mi was performed by permanently ligating the LAD. Therefore, a disturbed LAD distal blood flow is expected in the infarcted area until the neovascularization takes place to try compensate this blood flow reduction. Therefore, microbubble delivery in the ischaemic area will be delayed and probably reduced or even dramatically reduced since the infusion of the microbubbles takes place under continuous ultrasound imaging with concomitant destruction of the microbubbles.

Did the authors take into consideration a reduced delivery of Gal-3 shRNA in the iscaemic area in comparison to the border zone and non-infarcted areas and how do the authors comment on that? In case of a reduced Gal-3 shRNA delivery inside the ischaemic area, how do the authors explain the positive results obtained?

3.       In clinical settings, even in the first hours after MI a dyskinesia/hypokinesia/akinesia of the heart wall can be documented in echocardiographic measurements.

How do the authors explain their findings presented in Figure 4A depicting M-mode echocardiography images, where no evident hypokinesia/akinesia of the infarcted tissue can be seen 7days post MI, especially in the control group? However, this changes are seen at later timepoints such as 14 days and 21 days post MI.

Reviewer 2 Report

Overall, this paper is interesting and readable. I recommend publication after the authors have addressed the following comments.

Figure 2b – I can’t read the scale bar. I think it says 30 microns, which is very large compared to the DLS data shown. Adjust or add a zoomed in insert. The size distribution range in the micrographs looks larger than shown in the DLS graphs; it looks like the sub-micron particles were ignored. Please comment on the second, small-sized population that you’ve cut off in the DLS but that appears in the micrographs. Also for figure 2b – a graph with error bars for average DLS would be helpful to indicate the reproducibility of aggregate formation. This looks like a single measurement on one formation is being reported. Is the formulation reproducible?

The error bars in general are not explained. Duplicate experiments? Triplicate experiments? Triplicate measurements on the same sample? Different for different figures?

Figure 3b, why are both staining methods shown? It doesn’t appear that the information garnered is different with the two stains. If it is different, please explain. If it isn’t different, could the authors move one stained set to the supplemental information?

Abbreviations throughout the paper are hard to follow. US is never explicitly defined but is absolutely critical to the method. EF, FS, LVIDs, LVIDd are defined in a figure legend that occurs after the article text. Could the readers eliminate some of the abbreviations in favor of writing out the words if they only occur once or twice so that it is easier for the reader to understand/

This is a multidisciplinary project, so readers may not be experts in all areas. Add a brief description in section 3.3 to explain whether lower or higher values indicate better function. For example, what are we supposed to know after looking at figure 4a? If you’re a galectin expert broadly, you’ll be interested in this paper even if you don’t focus on heart disease, but you’ll struggle to understand the significance until the author flushes out this section and the discussion section pertaining to it.

In Figure 5a, the only reason the 7 day result is not significant and the 21 day result is significant is because the error bars are tighter on the 21 day graph. Therefore, an explanation of how the error bars were obtained is again quite important.

In section 4, it would be helpful to refer back to the figures that show the data from which the evaluations are being drawn.

In section 4.4, some template text was not deleted:

Authors should discuss the results and how they can be interpreted from the perspective of previous studies and of the working hypotheses. The findings and their implications should be discussed in the broadest context possible. Future research directions may also be highlighted

The conclusion should be more detailed. Restate what experiments were completed, and the % difference between results with control and galectin-3 microbubble. Are the differences significant enough to really say this “offers a new idea for the treatment of heart failure?” This may be too sweeping a conclusion once section 5 is expanded and may need to be reworded.

Reviewer 3 Report

Thank you for asking me to review this trial. This is undoubtedly an interesting trial, worthy of attention, aimed to evaluate the effectiveness of inhibition of Gal-3 in the development of myocardial fibrosis after myocardial infarction. Although the known limitation, the study is well designed and the preliminary results are newsworthy and surely future research directions could also be highlighted, as the authors reported.

Author Response

Point 1: Thank you for asking me to review this trial. This is undoubtedly an interesting trial, worthy of attention, aimed to evaluate the effectiveness of inhibition of Gal-3 in the development of myocardial fibrosis after myocardial infarction. Although the known limitation, the study is well designed and the preliminary results are newsworthy and surely future research directions could also be highlighted, as the authors reported.

Response 1: We sincerely thank the reviewer for your careful reading and positive feedback to our work. For the limitations mentioned in the paper, we will design corresponding experiments in the follow-up project to observe the duration of the Gal-3 inhibition effect and explore further mechanism of the anti-myocardial fibrosis by Gal-3 shRNA transfection.
